# Optical Camera Communication as an Enabling Technology for Microalgae Cultivation [note 1]

**DOI:** 10.3390/s21051621

**Published:** 2021-02-25

**Authors:** Cristo Jurado-Verdu, Victor Guerra, Vicente Matus, Carlos Almeida, Jose Rabadan

**Affiliations:** 1Institute for Technological Development and Innovation in Communications (IDeTIC), Universidad de Las Palmas de Gran Canaria (ULPGC), 35017 Las Palmas de Gran Canaria, Canary Islands, Spain; vguerra@idetic.eu (V.G.); vmatus@idetic.eu (V.M.); jrabadan@idetic.eu (J.R.); 2Spanish Bank of algae (BEA), Instituto de Oceanografía y Cambio Global (IOCAG), Fundación Canaria Parque Científico y Tecnológico, Universidad de Las Palmas de Gran Canaria (ULPGC), 35230 Las Palmas de Gran Canaria, Canary Islands, Spain; carlos.almeida@ulpgc.es

**Keywords:** optical camera communications, visible light communications, microalgae cultivation, artificial lighting, light management, smart farming, Agriculture 4.0

## Abstract

Optical Camera Communication (OCC) systems have a potential application in microalgae production plants. In this work, a proof-of-concept prototype consisting of an artificial lighting photobioreactor is proposed. This reactor optimises the culture’s photosynthetic efficiency while transmitting on-off keying signals to a rolling-shutter camera. Upon reception, both signal decoding and biomass concentration sensing are performed simultaneously using image processing techniques. Moreover, the communication channel’s theoretical modelling, the data rate system’s performance, and the plant distribution requirements and restrictions for a production-scale facility are detailed. A case study is conducted to classify three different node arrangements in a real facility, considering node visibility, channel capacity, and space exploitation. Finally, several experiments comprising radiance evaluation and Signal-to-Noise Ratio (SNR) computation are performed at different angles of view in both indoor and outdoor environments. It is observed that the Lambertian-like emission patterns are affected by increasing concentrations, reducing the effective emission angles. Furthermore, significant differences in the SNR, up to 20 dB, perceived along the illuminated surface (centre versus border), gradually reduce as light is affected by greater dispersion. The experimental analysis in terms of scattering and selective wavelength attenuation for green (*Arthrospira platensis*) and brown (*Rhodosorus marinus*) microalgae species determines that the selected strain must be considered in the development of this system.

## 1. Introduction

Microalgae culture has gained significant momentum during the last decade, triggered by the necessity of developing new and sustainable resources. They have become a promising alternative source for biofuels and biogas production, human and animal nutrition, cosmetics and bioactive supply for nutraceutical and pharmaceutical applications.

Microalgae biomass production is currently carried out using both open ponds (e.g., raceways) and closed photobioreactors [1]. The last ones are preferable at the laboratory and pilot-plant scales since, with the appropriate design, they can optimise growth conditions (nutrient levels, carbon concentration, temperature, acidity, among others). Among these parameters, light radiant energy is a capital factor that affects the photosynthetic efficiency, and therefore the overall productivity [2].

Generally, microalgae production plants are designed to take advantage of the Sun as the primary light source due to cost optimisation. However, the Sun’s irradiance depends on several factors, such as weather conditions, latitude and day time. Furthermore, these open-air plants need vast extensions to be profitable since effective biomass production depends on the directly-exposed surface. Nonetheless, artificial lighting can provide advantages in photosynthetic efficiency (custom spectrum and intensity profiles) and the tight control they offer concerning microalgal biochemistry and growth, increasing industrial processes’ reliability. Using this type of lighting, biomass production depends no longer on the plant’s exposed surface, but on its equivalent volume. Although optimising light quality could considerably reduce energy consumption, another approach concerns how light is delivered to the culture: flashing (pulsed) light instead of continuous illumination.

In this context, pulsed illumination could serve as a communication link, enabling the re-utilisation of light sources as effective visible light communication (VLC) transmitters [3]. The low bandwidth transmission channel they provide (attaining the stated frequency restrictions) could be used for online monitoring of the culture’s state conditions via deployable sensors within the photobioreactor module.

Nevertheless, there are still a few parameters that are either too complex to be measured in situ or imply invasive methods. These parameters are microalgae biomass concentration and its growth phase. In [4,5], different approaches based on the digital processing analysis of red-green-blue (RGB) images for low cost, fast and accurate quantification of biomass concentration were proposed and experimentally validated. Therefore, cameras could be used as sensing devices for these parameters.

On the other hand, the use of cameras as communication receivers for VLC links is a research topic that is receiving significant attention. Furthermore, there is a current standard specification by the working group IEEE 802.15.7 [6], which has finally integrated this new strategy known as optical camera communication (OCC) [7,8,9,10]. These devices are bandwidth-limited compared to traditional photodetectors, such as p-type, intrinsic, n-type (PIN) and avalanche photodiodes (PDs). However, due to the use of image-forming optics [11], cameras can receive light from multiple sources, providing inherent spatial multiplexing capabilities [12,13,14], which can be easily exploited for simultaneous monitoring within a microalgae production plant.

This work proposes the use of low-cost cameras for both remotely sensing microalgae culture parameters and for establishing a direct communication link with flat-panel photobioreactors. Compared to other radio or wired technologies, the use of OCC has the following advantages. First of all, this technology reuses the light that comes from the photobioreactors and makes better use of this excess of energy that would otherwise be wasted. Second, it replaces a generic communications coordinator with an intelligent camera that performs additional routines apart from establishing a communication link. On the one hand, it carries out continuous surveillance of the reactors and the personnel who access the room and can act as an early warning system. On the other hand, it simplifies the estimation of some culture parameters that are difficult to measure or involve intrusive procedures, such as the concentration of biomass or the strain’s growth state. For example, for evaluating the biomass concentration, high-cost equipment must be used to automate the extraction of representative samples. In the worst case, this procedure must be carried out periodically by the staff. Furthermore, concerning communications security, an extra layer of protection is added thanks to the light confinement within the room. Finally, cameras’ inherent spatial multiplexing capabilities significantly simplify link protocols, making communications more robust and less prone to errors.

This work presents part of the ATICCuA Project results, a multidisciplinary research project carried out by the Spanish Bank of Algae (BEA by its Spanish acronym ) and the Photonics and Communications Division of the Institute for Technology Development and Innovation in Communications (IDeTIC). This project addresses the development of prototypes based on the visible light communication (VLC) and, more specifically, those used in underwater wireless optical communications (UWOC) for application in microalgae culture systems. Precisely, its main objective consists of the design of an LED-based dual-use system, which provides configurable lighting for the culture and production of microalgae and cyanobacteria and optical wireless communication capabilities for optical underwater channel characterisation.

This work introduces and discusses the microalgae cultivation restrictions that affect the communications’ performance and the overall system’s design. In addition, a comprehensive analysis based on geometrical constraints is carried out, providing results such as optimal camera positioning concerning a custom-defined metric that relates the aggregate data rate and effective space exploitation. From this analysis, a preliminary study of the plant distribution of a case study is carried out. Furthermore, several experiments that evaluate the system’s communications performance are conducted in an indoor and outdoor scenario.

The remainder of the paper is structured as follows. Section 2 introduces the proposed OCC-based architecture in a bottom-up manner. Section 3 groups together the discussion of the channel model, the analysis of the data rate achievable for each container and its optimal distribution in the plant for the efficient deployment of this technology. Section 4 describes the methods, materials and procedures involved, on the one hand, in the study of the plant distribution for a real application case and, on the other hand, in the two experiments conducted for the evaluation of the prototype based on the quality of the optical signal received by the camera. Section 5 presents the results obtained, their interpretation and a discussion. The conclusions of this work are summarised in Section 6.

## 2. Proposed Architecture

The proposed architecture follows a many-to-one unidirectional network topology where photobioreactor nodes transmit sensor-related data to a receiver camera node (Figure 1).

The photobioreactor nodes comprise a uniform-radiance LED flat panel attached to a glass container that holds the microalgae culture. The receiver node consists of a rolling-shutter (RS) camera connected to the communication endpoint that exposes a control interface and a standard wired communication bus. The transmitter nodes use two different signals to establish a link: a well-known beacon signal that uniquely identifies the node and data packets composed of Manchester-encoded pulses. The receiver uses the beacon signal to perform three separate routines: establishing the communication link, estimating the channel response to complete the decoder training, and finally, sensing the culture by estimating the biomass concentration and growth phase of the microalgae species. Therefore, this system performs two functions. It establishes a monitoring link and directly senses essential culture parameters.

In the following sections, both nodes are presented, highlighting the influence of the microalgae cultivation needs and requirements in their design.

### 2.1. Photobioreactor Node

The transmitter is comprised two parts: the LED flat panel and the LED driver that generates the corresponding OCC driving signal to transmit sensor data provided by probes deployed inside the container (such as temperature, acidity, carbon and nutrient levels). This section is focused on light generation and transmission.

As was previously mentioned, the competitiveness of any artificial light-driven microalgal production requires improvements in photon harvesting and the conversion efficiency of light sources [2]. Hence, it is crucial to optimise light quality and delivery. In terms of light energy, the photosynthetic rate is directly related to the irradiance power, and excessive or insufficient incident light constrains optimal performance and may induce the photo-inhibition and photo-oxidation of the cells, eventually attaining photo-damage and even leading to culture death [15]. In terms of light spectra, the radiant energy absorbed by microalgae highly depends on the chemical nature of their native pigments, which have specific absorption bands in the visible and near-infrared spectra. Thus, better energy usage can be achieved by adjusting the light source’s emission to match the absorption spectrum [16].

On the other hand, recent experimental studies have shown that combining short and intense light flashes with extended dark periods instead of continuous illumination might increase the culture’s growth efficiency, as discussed in [17,18]. The light frequency and duty cycle are particular to each microalgae species and the expected biomass product result (lipids, carotenoids, etc.). It may vary between a few Hz up to tens of kHz.

Therefore, the transmitting source’s design must consider all these restrictions: light quality (intensity profile, spectra) and light delivery (frequency and duty cycles) concerning the selected strain and the expected results.

### 2.2. Camera Node

The acquisition mechanism of image sensors inherently limits the available bandwidth for communications. In global-shutter (GS) cameras, the whole image sensor is exposed simultaneously. Thus, the achievable data rate is upper-bounded by their frame rate, restraining the light source’s switching frequency considerably. On the other hand, RS cameras scan the scene sequentially row by row of pixels (usually on the shorter dimension of the sensor), allowing capturing different light states (intensity, colour variations, among others) within the lamp’s source projection in the frame [19]. In this case, the theoretical bandwidth limit is imposed by the row shift time, commonly denominated the sampling time, $t_{s}$, which is the fixed duration between the start of a row scan and the consecutive one. Furthermore, these rows are not disjointedly exposed, but in an overlapping manner. This overlap duration depends on the configured exposure time, $t_{\exp}$, which is the span of time during which each row of pixels is integrating light. In terms of communications, the effect of this overlap can be modelled (for a uniform light source) as the product of a weighted moving-average filter, with its corresponding transfer function Equation (Equation 1), which further restricts the effective bandwidth (cut-off frequency—Equation (Equation 2)).
(1)$$H\left(w\right)=\frac{e^{-jw/2(N-1)}}{N}\frac{sin(wN/2)}{sin(w/2)}$$
where *N* is an integer that relates the exposure time to the sampling time ($N=t_{\exp}/t_{s}$) and w is the normalized angular frequency in radians per sample ($w=2\pi f/f_{s}$).

The cut-off frequency, $w_{3dB}$, defined as the half-power point’s frequency, can be computed using the modulus squared function of the transfer function using Equation (Equation 2).
(2)$${\left|H\left(w_{3dB}\right)\right|}^{2}=\frac{1}{N^{2}}\frac{sin^{2}\left(\frac{w_{3dB}N}{2}\right)}{sin^{2}\left(\frac{w_{3dB}}{2}\right)}=\frac{1}{2}$$

Equation (Equation 2) does not have a general analytical solution. However, it is possible to rely on numerical methods such as Newton–Raphson’s algorithm to determine the cut-off frequency that sets the system’s available bandwidth (Equation (Equation 3)).
(3)$$f_{tx}\leq f3\mathrm{dB}/2$$

Therefore, to increase the effective transmission bandwidth, it is necessary to minimise the camera’s exposure time. Consequently, the PDs of the pixels exposed for a shorter time have a lower signal-to-noise ratio (SNR). In previous works, it was shown that the reduced received power due to the extinction along the path or due to low exposure times can be overcome using the analogue amplifier of the CMOS camera by increasing the gain it provides before the analogue-to-digital converter (ADC), ultimately improving the SNR [20,21].

## 3. Communications Modelling

The proposed architecture presents some specific characteristics from the communication system point of view. First, the light propagates through different media from the emitter to the camera, which modifies the received optical signal. On the other hand, the camera position affects the communications’ performance since it determines the received power and the emitter image size. Furthermore, the distribution and size of the photobioreactors also impact the complete OCC system performance. In this section, all these aspects are addressed.

### 3.1. Communication Channel

The channel can be divided into different layers. Figure 2 details the layered version of the channel, which is composed of: the inner air gap layer, the inner glass layer, the microalgae suspension in the water layer and the outer glass layer that emits the light that reaches the camera. This work focuses on the analysis of these four primary layers of the channel. The effect of the link’s air gap between the container and the camera was addressed previously in [22].

This preliminary analysis shows two critical interfaces: the inner glass/water and the outer glass/air interfaces that affect communications’ performance and constrain harvesting optimisation.

Light rays emitted in a specific direction by the light source ($\vartheta_{0},\varphi_{0}$) change their trajectory upon reaching the surface of the inner glass ($\theta_{\mathrm{ig}},\varphi_{\mathrm{ig}}$). Then, those rays reach the first critical interface (inner glass/water interface) and undergo partial or total internal reflection depending on the critical angle ($\Theta_{\mathrm{iglass}}/\mathrm{water}=51.9^{\circ }$) ( considering refraction indices of $n_{air}=1.0$, $n_{glass}=1.69$ and $n_{water}=1.33$ for the air, glass and water media, respectively). However, total internal reflection will never happen at this critical interface in the considered conditions. The reason is that as soon as $\theta_{0}$ approaches 90 degrees, $\theta_{\mathrm{ig}}$ tends to 31${}^{\circ }$, which is considerably lower than $\Theta_{\mathrm{glass}}/\mathrm{water}$. In the microalgae medium, light rays travel with direction ($\vartheta_{\mathrm{water}},\varphi_{\mathrm{water}}$). Those rays that reach the top, bottom, left or right side reflectors of the container are reflected. Figure 2 shows an example of one ray reaching the container’s right side and being reflected.

Finally, within the outer glass, light propagates with ($\vartheta_{\mathrm{og}}$, $\varphi_{\mathrm{og}}$) until it arrives at the second critical interface. The outer glass-air interface has its corresponding critical angle ($\Theta_{\mathrm{glass}}/\mathrm{air}=36.28^{\circ }$). Light rays with incident angles greater than $\Theta_{\mathrm{glass}}/\mathrm{air}$ are reflected, reducing the total optical power leaving the photobioreactor.

This preliminary analysis aims to better understand the behaviour of the radiance *L*($\vec{r},\vec{\theta }$) of the extended outer glass surface (Equation (Equation 4)); in other words, the radiant intensity ϕ, emitted from an infinitesimal unit surface, $d_{A_{⊥}}$, and contained within a unit of solid angle aligned normal to the direction of interest, $\vec{\theta }=\left(\vartheta ,\varphi \right)$, for a particular location, $\vec{r}=\left(x,y\right)$.
(4)$$L\left(x,y,z=z_{0}\vartheta ,\varphi \right)=\frac{d^{2}\phi \left(x,y,z=z_{0},\vartheta ,\varphi \right)}{d\Omega dAcos\vartheta }$$

After characterising the radiance of the surface, the total irradiance over a pixel is obtained by integrating the incoming radiance from any direction in the normal hemisphere that encloses the pixel area Equation (Equation 5).

As an example, Figure 3 shows the irradiance per unit of area of an infinitesimal portion, $d_{V}$, of a virtual surface that encloses the container keeping the same horizontal distance to the container’s centre point.
(5)$$E\left(x,y,z\right)=\int_{\Omega }L\left(x,y,z=z_{0},\vartheta ,\varphi \right)cos\vartheta d\Omega$$

In the stated conditions, the radiance would be affected by different phenomena: the culture’s absorption capacity, the shading effect between cells, and light scattering. When the amount of biomass is negligible compared to the volume of water in the container, these phenomena might be neglected, and radiance can be estimated using geometrical optics. Now, considering that the channel consists of symmetrically repeating parallel layers that start and end in the same air medium, light’s incident angle $\vartheta_{0},\varphi_{0}$ coincides with exiting radiance angle $\vartheta_{\mathrm{out}},\varphi_{\mathrm{out}}$ without any hard restriction as there is no total internal reflection. Figure 2 shows light example paths in 2D dimensions.

Now, considering that the bottom, top, left and right container’s sides reflect light in a specular manner, the radiance of a point ($x^{\prime }$, $y^{\prime }$, z=0) of the external surface in any direction $\vec{\vartheta^{\prime }}$ is a fraction of the radiance emitted by the light source in the same direction (or in a shifted direction, if the ray comes from a reflection in the walls) ($\vec{\vartheta^{\prime }}$), but from a translated origin point, *x*,*y*, of the output point ($x=x^{\prime }+a$,$y=y^{\prime }+b$).

As a result, the output radiance from infinitesimal surfaces located at the container’s centre tends to mimic the original lamp radiation pattern. However, the output radiance at the borders of the container tends to skew. This skewness occurs because there are no direct light contributions. This skewness effect of the radiance pattern significantly constrains the communications’ performance, as the optical signal power recovered from specific viewpoints would not be enough to establish effective communication. As it can be extracted from preliminary experiments, this skewness can be partially reduced if the container’s sides reflection has a perfect diffuse behaviour, causing the light to be uniformly spread in all directions. However, the skewness cannot be fully mitigated since the light rays that contribute to reducing this effect come from very steep entry angles and repeatedly bounce within the container boundaries, ultimately reducing its optical power.

Regarding light power exiting the surface, it is important to consider the fraction of the incident light reflected at each interface. Considering non-polarised light the effective reflection coefficient, for each input angle, $\vartheta_{i}$, can be expressed by Equation (Equation 6).
(6)$$R_{\mathrm{eff}}=\frac{\left(R_{s}+R_{p}\right)}{2}$$
where $R_{s}$ (Equation (Equation 7)) and $R_{p}$ (Equation (Equation 8)) are the reflectances for s-polarized and p-polarized light, respectively.
(7)$$R_{s}=|\frac{n_{1}\cos\vartheta_{i}-n_{2}\sqrt{1-{\left(n_{1}/n_{2}\;\cdot \;\sin\vartheta_{i}\right)}^{2}}}{n_{1}\cos\vartheta_{i}+n_{2}\sqrt{1-{\left(n_{1}/n_{2}\;\cdot \;\sin\vartheta_{i}\right)}^{2}}}{|}^{2}$$
(8)$$R_{s}=|\frac{n_{1}\sqrt{1-{\left(n_{1}/n_{2}\;\cdot \;\sin\vartheta_{i}\right)}^{2}}-n_{2}\cos\vartheta_{i}}{n_{1}\sqrt{1-{\left(n_{1}/n_{2}\;\cdot \;\sin\vartheta_{i}\right)}^{2}}+n_{2}\cos\vartheta_{i}}{|}^{2}$$
where $n_{1}$ and $n_{2}$ are the refractive indexes of both mediums. Hence, light rays with acute entry angles have a significant amount of power reflected. This has some implications. At first, not all the optical power reaches the microalgae culture, and it will depend on the radiance pattern of the surface and the effective emission angle of the source. Finally, not all the optical power leaves the container’s surface, which is not desirable for establishing an optical communication link despite being suitable for cultivation. Under this initial configuration, when the biomass concentration is above a threshold level, the scattering and the absorption phenomena cannot be neglected. In that case, Beer–Lambert’s law can be used to describe the attenuation of light due to absorption by the biomass concentration (Equation (Equation 9)). This equation states that the attenuation of light over a distance is proportional to the light intensity, where *C* is the volumetric absorption coefficient. The latter is the product of the specific light absorption coefficient, $\alpha_{x,\lambda }$, and the biomass concentration (ρ). The integration of Equation (Equation 9) over the light path, taking into account the wavelength dependency, results in Equation (Equation 10).
(9)$$\begin{array}{cc} \frac{dI_{\lambda }\left(x\right)}{d(x)} & =-C_{x,\lambda }\;\cdot \;I=-\alpha_{x,\lambda }\;\cdot \;\rho \;\cdot \;I \end{array}$$
(10)$$\begin{array}{cc} I(x) & =\sum_{\lambda =800}^{\lambda =300}I_{\lambda }\left(0\right)\;\cdot \;e^{-\alpha_{z,\lambda }\;\cdot \;\rho \;\cdot \;x} \end{array}$$

However, as light travels through the photobioreactor, it is absorbed and scattered by the microalgae. The light intensity $I_{\lambda }\left(r,\vec{d}\right)$ (Wm${}^{-2}$sr${}^{-1}$) at a given location within the container, *r*, and in the direction $\vec{s}$ can be determined by solving the radiative transfer equation (RTE) [23], which represents an energy balance on the radiative energy travelling along a particular direction. In steady-state conditions, the RTE for non-collimated light, as in this case, can be expressed as Equation (Equation 11).
(11)$$\vec{s}\;\cdot \;\nabla I_{\lambda }\left(r,\vec{d}\right)=-\beta_{\lambda }I_{\lambda }\left(r,\vec{d}\right)+\frac{\sigma_{\lambda }}{4\pi }\int_{4\pi }I_{\lambda }\left(r,\vec{d}\right)\;\cdot \;\Phi_{\lambda }\left(\vec{s_{i}}\to \vec{s}\right)d\Omega$$
where $\Phi_{\lambda }$ is the scattering phase function (SPF), which represents the angular distribution of the scattered light, in other words, the probability that radiation travelling in a given direction, $\vec{s_{i}}$, will be scattered to the direction, $\vec{s}$, of interest. This function is determined by the size, shape and refractive index distribution of the scattering particle. $\beta_{\lambda }$ is the extinction coefficient (m${}^{-1}$) composed by the scattering coefficient, $\sigma_{\lambda }$, and the absorption coefficient, $\kappa_{\lambda }$ (m${}^{-1}$). These two parameters can be expressed in terms of the averaged scattering $C_{\mathrm{sca}}$ and absorption $C_{\mathrm{abs}}$ cross-sections (m${}^{2}$), respectively (Equation (Equation 12)).
(12)$$\sigma_{\lambda }=C_{\mathrm{sca}}\;\cdot \;N\;\mathrm{and}\;\kappa_{\lambda }=C_{\mathrm{abs}}\;\cdot \;N$$
where *N* is the microorganism concentration expressed in number of cells per m${}^{3}$ of water suspension.

This RTE equation given by Equation (Equation 11) reveals that the scattering absorption cross-sections and the SPF have an important role in predicting light transfer in photobioreactors for simulation, design and optimisation purposes. However, these characteristics are interrelated and difficult to estimate from the electromagnetic wave perspective, given the microorganisms’ complex morphology. Nonetheless, they can still be measured experimentally with more or less difficulty and related to some parameters of the cell, as detailed in [24]. Furthermore, these radiative properties of microalgae (the absorption cross-section, scattering cross-section and scattering phase function) vary significantly over time, depending on the strain’s growth stage, as shown by the studies [25,26]. As all these radiative parameters depend on the wavelength of light and because they vary in such a significant way through time, it can be considered that the values obtained by an RGB camera would correctly identify not only the strain, but its growth’s state.

Regarding the communications’ performance, although the presence of these microorganisms attenuates the signal before it reaches the exposed container’s surface, it may be beneficial in some cases because it distributes the optical power more evenly, uniforming the radiation pattern over the entire container’s surface. In conclusion, microalgae particles’ scattering phenomena can be exploited for communications in scenarios where the camera is not facing perfectly perpendicular to the transmitter surface.

### 3.2. Data Rate Analysis

The maximum achievable data rate for a photobioreactor depends on its projection’s vertical size over the image; in other words, the total number of vertical samples (rows of pixels) recovered from the signal [27]. Consequently, it is necessary to consider the scenario’s geometrical configuration, the relative positions between the camera and the photobioreactors and their size.

Equations (Equation 13) and (Equation 14) relate the projection dimensions (in meters) of a rectangular surface over the scene plane, $H_{p}$, $W_{p}$, with its corresponding pixels dimensions, $h_{roi}$, $w_{roi}$. Figure 4 shows all the geometry and camera parameters involved in a generalised scenario.
(13)$$\begin{array}{cc} h_{roi} & =H_{p}\;\cdot \;\frac{h_{\mathrm{res}}}{2\;\cdot \;\tan\left({\mathrm{AoV}}_{V}/2\right)\;\cdot \;D_{\mathrm{FoV}}}\;\left(pixels\right) \end{array}$$
(14)$$\begin{array}{cc} w_{roi} & =W_{p}\;\cdot \;\frac{w_{\mathrm{res}}}{2\;\cdot \;\tan\left({\mathrm{AoV}}_{H}/2\right)\;\cdot \;D_{\mathrm{FoV}}}\;\left(pixels\right) \end{array}$$
where $h_{res}$ and $w_{res}$ are the pixel dimensions of the image, $AoV_{h}$ and $AoV_{w}$ the horizontal and vertical angle of view (AoV) of the camera (in degrees), respectively, and $D_{FoV}$ the distance (in meters) from the camera lens to the scene plane.

Following Nyquist’s criterion ($f_{\mathrm{tx}}\leq f_{3dB}/2$), the minimum number of vertical samples of the signal required per symbol can be computed using Equation (Equation 15).
(15)$$h_{s}=\frac{f_{s}}{f_{\mathrm{tx}}}\leq \frac{2}{f_{3dB}\;\cdot \;t_{s}}\;\left(\mathrm{samples}\right)$$

It must be remarked that communication takes place in a windowed manner. Leaving aside the reflections with the objects present in the scene [13,28], the signal has to be recovered mainly from the projection within the image of the light source where the signal quality is considerably better. Hence, during the acquisition, only a fraction of the total data sent is sampled by the sensor. While the sensor is not scanning the transmitter’s surface, but another part of the scene, it will remain blind to the light changes of the transmission [19]. To overcome these blind periods, the data packets must be sent repeatedly (at least while the camera is acquiring two full frames). In addition, to avoid packet losses, the source’s image projection, $h_{p}$, must fit at least two complete packets, as was detailed in the previous work [29]. With these restrictions (Equation (Equation 17)), the overall transfer rate (in bauds) is obtained from Equation (Equation 16).
(16)$$\begin{array}{cc} R_{b} & =S_{packet}\;\cdot \;\frac{\mathrm{fps}}{2} \end{array}$$
(17)$$\begin{array}{cc} h_{\mathrm{packet}} & \geq 2\;\cdot \;S_{\mathrm{packet}}\;\cdot \;h_{s} \end{array}$$
where $S_{\mathrm{packet}}$ is the equivalent number of symbols per data packet. On the other hand, the projection’s width in pixels (number of columns) also plays an important role in communications. Pixels located in the same row are exposed to light simultaneously, and they can be used to filter out noise, thus strengthening the signal-to-noise ratio. Moreover, wider areas aid source discovery and tracking and considerably ease the decoding routine. As a consequence, a minimum pixel width must be selected as a design requirement.

Finally, it is worth mentioning that the proposed system’s achievable capacity considerably exceeds the requirements for accurate monitoring of the culture parameters. To obtain a preliminary idea of these requirements, the following considerations are taken into account. The selected parameters to be measured are acidity, temperature, light conditions, carbon, nutrients, inhibitors presence and O${}_{2}$ degassing. The packet’s payload allocates twenty bytes per parameter, which is a much larger allocation than necessary. Furthermore, given that the culture’s temporal evolution is considerably slow, samples of the culture can be taken reliably every 5 min, which offers a substantially acceptable temporal resolution. Based on these considerations, the required capacity is about four bits per second. Therefore, the hundreds of bauds per second that can be sent using the proposed system, as specified in Section 4.1, satisfy the stated capacity requirements.

### 3.3. Plant Distribution

The distribution of the photobioreactor nodes across the room plays an essential role in communications. To establish a link, the receiver should visualise each node, and consequently, it is necessary to reserve some space free of obstructive interference between the transmitters and the camera. This reveals the importance of analysing different possible solutions for placing the nodes until finding the one that best suits the project’s initial requirements, either in terms of better link quality, higher capacity or more efficient space exploitation.

In a previous work [22], a metric was proposed to compare the performance of different solutions. Nevertheless, the complex nature of this problem, which involves several variables and a wide range of possible initial requirements, highlighted the need to modify this classification strategy. This work proposes replacing the original metric based on a single value with a modified multidimensional metric, Equation (Equation 18), allowing a more flexible classification of the arrangements. This new metric offers a comparison tool in which it is left to the designer’s discretion to select a specific application design’s priorities.
(18)F:{N,Q,SUR}
(19)$$\mathrm{where}:\;\mathrm{SUR}=\frac{V_{\mathrm{cont}}}{V_{\mathrm{room}}}$$

It consists of three independent variables: the number of simultaneous photobioreactors monitored by a single camera, *N*, a communications performance metric, *Q*, and the space utilisation ratio (SUR). The first two variables account for the overall achievable data rate of the setup. The last term relates the total equivalent volume of the containers bound to biomass harvesting, $V_{\mathrm{cont}}$, and the minimum room volume needed, $V_{\mathrm{room}}$. The optimisation of this term has direct implications in the reduction of production costs and, therefore, the viability and competitiveness of cultivation plants based on artificial lightning and vertical racks.

The proposed *Q* metric is derived from the well-known Shannon–Hartley’s equation [30] for estimating the channel capacity for each container (Equation (Equation 20)).
(20)$$C=BW\;\cdot \;log_{2}\left(1+S/N\right)$$
where *C* is the capacity in bits per second, BW is the bandwidth of the channel in Hertz and S/N is the SNR, expressed as a linear power ratio. As was previously mentioned, communications happen in a windowed manner. In other words, the transmission effectively takes place during a fraction of the time to acquire a frame, $t_{\mathrm{frame}}$. This fraction of time, or channel availability, $\tau_{\mathrm{rx}}$, depends on the geometrical configuration, $t_{\mathrm{geo}}$, and on the elapsed time between when the camera finishes capturing one frame and starts with the next one $t_{\mathrm{inter}}$ (Equation (Equation 22)). However, this last term is considerably lower than $t_{\mathrm{frame}}$ and can be neglected.

Adding this factor to Equation (Equation 20), the effective capacity is obtained for each container, as Equation (Equation 21) shows.
(21)$$C=BW\;\cdot \;log_{2}\left(1+S/N\right)\;\cdot \;\tau_{\mathrm{rx}}$$
(22)$$\mathrm{where}:\;\tau_{\mathrm{rx}}=\frac{t_{\mathrm{geo}}}{t_{\mathrm{frame}}+t_{\mathrm{inter}}}\approx \frac{h_{roi}}{h_{res}}$$

The proposed metric *Q* is then defined as the relationship between the channel’s capacity for a particular arrangement and container, *C*, and the ideal capacity, $C_{ideal}$, Equation (Equation 23).
(23)$$Q=\frac{C}{C_{\mathrm{ideal}}}=\frac{BW}{BW_{\mathrm{ideal}}}\;\cdot \;\frac{log_{2}\left(1+S/N\right)}{log_{2}\left(1+{\left(S/N\right)}_{\mathrm{ideal}}\right)}\;\cdot \;\frac{\tau_{\mathrm{rx}}}{\tau_{\mathrm{rx}},\mathrm{ideal}}$$

The vast number of parameters involved in this metric’s computation makes this metric unreasonable for a practical analysis of different plant distributions, even more so if there are plenty of configurations to be compared. Hence, for this metric to be a useful tool for this purpose, it is necessary to assume a series of coarse approximations. These approximations would greatly simplify the practical comparison of two cases without incurring harsh penalties. First, it is important to remark that the camera’s hardware and configuration would not change. Consequently, the bandwidth, which depends on the sampling frequency and the image sensor’s exposure time, remains constant ($BW^{\prime }=BW_{\mathrm{ideal}}$). On the other hand, the maximum capacity will be achieved when the transmitting source occupies the image entirely in the scanning dimension (usually from top to bottom). In this way, the link availability is kept while capturing a frame ($\tau_{\mathrm{rx}},\mathrm{ideal}=1$). With these considerations, Equation (Equation 23) could be reduced to Equation (Equation 24).
(24)$$Q=\frac{log_{2}\left(1+S/N\right)}{log_{2}\left(1+{\left(S/N\right)}_{\mathrm{ideal}}\right)}\;\cdot \;\tau_{\mathrm{rx}}$$

Now, using the first order Taylor approximation of the logarithmic function (at x = 0), Equation (Equation 25), results in Q as given by Equation (Equation 26).
(25)$$\lim_{x\to 0}log_{2}\left(1+x\right)=\frac{x}{ln(2)}$$
(26)$$Q\approx \frac{S/N}{{\left(S/N\right)}_{\mathrm{ideal}}}\;\cdot \;\frac{h_{\mathrm{roi}}}{h_{\mathrm{res}}}$$

In this way, the measurement of the capacity of a placement can be approximated to the comparison of the SNR with the best case and the size of its projection in the image. However, assumptions regarding the relationship between the SNR under study and the best-case SNR can simplify the analysis. These assumptions are detailed below. First, the container radiance varies smoothly through the entire surface, or at least in the area were the signal is recovered. Not the whole surface of the container is utilised for signal detection, but a fraction of it. The radiance at any given direction is expressed with respect to the maximum radiance $L_{\max}$ (Equation (Equation 27)).
(27)$$L\left(\vartheta ,\varphi \right)=L_{\max}\hat{L}\left(\vartheta ,\varphi \right)$$

The pixel FoV is small enough to assume that the pixel irradiance’s contributions come from the same emitter radiance’s output angles. Furthermore, the use of image-forming optics compensates the power loss due to spherical propagation with the projected size of the optical source on the image sensor [31]. Therefore, the power received by a pixel (Equation (Equation 28)) in a given direction can be approximated as the ratio of the total received power in the direction of maximum radiation, and it can be related to the ratio of the emitted radiance.
(28)$$S_{\mathrm{pixel}}\left(\vartheta ,\varphi \right)=S_{\mathrm{pixel}},\max\;\cdot \;\hat{L}\left(\vartheta ,\varphi \right)$$

Furthermore, assuming that the region of interest (ROI) where the signal will be recovered is constant in width and height, then the total received power would be the aggregation of all contributions from the ROI’s pixels. Therefore, the optical power received within the ROI is approximated by Equation (Equation 29).
(29)$$S_{\mathrm{roi}}\left(\vartheta ,\varphi \right)=S_{\mathrm{roi}},\max\;\cdot \;\hat{L}\left(\vartheta ,\varphi \right)$$

Regarding the noise power, no external interfering optical sources were considered for simplicity. The primary noise sources are thermal noise ($\sigma_{\mathrm{th}}$), shot noise ($\sigma_{\mathrm{shot}}$) and quantisation noise ($\sigma_{\mathrm{adc}}$), which is generated by the ADC of the camera. Besides, thermal noise does not depend on the signal power, whilst shot noise is affected by the received optical power. Nonetheless, considering the application scenarios of this work, it is expected that the shot noise contribution could be neglected with respect to thermal noise. Hence, the noise power is the same regardless of the receiver’s position.

Therefore, comparing the radiation at different viewing angles can give a good approximated idea about the difference in the SNR of different container’s arrangements.
(30)$$\frac{\left(S/N\right)}{{\left(S/N\right)}_{\mathrm{ideal}}}\approx \hat{L}\left(\vartheta ,\varphi \right)$$
(31)$$\mathrm{where}:\;\hat{L}\left(\vartheta ,\varphi \right)=\frac{L(\vartheta ,\varphi )}{\max\{L(\vartheta ,\varphi )\}}$$

In conclusion, the final approximation of Q (Equation (Equation 32)) is obtained by combining Equations (Equation 24) and (Equation 30).
(32)$$Q\approx \hat{L}\left(\vartheta ,\varphi \right)\;\cdot \;\tau_{\mathrm{rx}}$$

As mentioned, this metric is intended to be easy to compute and practical when guiding the engineer to distribute the plant most optimally. The approximations stated above greatly simplify the comparison of two cases without incurring non-affordable penalties. This Q parameter allows indirectly approximating the maximum transmission rate achievable by each photobioreactor node (Equation (Equation 17)).

In this work, the minimum value of Q (${Q|}^{\min}=min\left\{Q_{c,1},Q_{c,2},...,Q_{c,N}\right\}$), of all the containers, is used to evaluate a particular arrangement. The reason is that as an initial requirement, all the containers must share the same data rate. To achieve this, all the emitters will adapt their transmission to the minimum available in the scene. It should be clarified that this metric varies between zero and one, where one implies the maximum theoretical performance (optimised use of channel capacity).

## 4. Methodology

In this work, a preliminary analysis of the plant distribution of a case study was carried out. In addition, several experiments to evaluate the system’s signal reception in different scenarios were conducted. In the following sections, these experiments are described separately following the scheme: description, materials and resources, methods and data analysis.

### 4.1. Plant Distribution Study

This section describes the preliminary study of the best plant distribution for a real case study. In this case, it is intended to locate 12 custom containers of 150 × 50 × 9 cm (height by width by depth) in a 300 × 200 × 300 cm empty room. These containers must be placed in individual racks capable of holding up to 4 containers, two at the top and two at the bottom.

In terms of communication performance, those cases must satisfy the following derived constraints. The top photobioreactors will fill the upper half of the image and the lower ones the lower half. Consequently, the camera should always point to the shelf’s vertical centre, and it will be aligned with respect to the centre (in the vertical dimension). In this way, the transfer rate is ensured to be equal for each reactor, with the number of vertical pixels equal to half of the image’s vertical resolution (Equation (Equation 33)). The minimum pixel width of any reactor was set to 30 pixels (Equation (Equation 34)). This value was selected after conducting preliminary experimental tests. Finally, it must be highlighted that the sensor’s aspect ratio relates both the vertical and the horizontal FoV, ${\mathrm{aspect}}_{\mathrm{ratio}}$; thus, altering one of them will influence the other.

To properly address this analysis, it is necessary to select a camera as an example. The camera selected is the PiCamera v2, detailed in Table 1. Using this camera with an exposure time of 57 s and attending to the previously stated conditions, the maximum baud rate achievable is approximately 717 (Bd/reactor) per available channel. Considering three independent communication channels, red, green and blue, the maximum baud rate is 2151 (Bd/reactor). This value is obtained by combining Equations (Equation 15), (Equation 16), (Equation 17) and (Equation 33) into Equation (Equation 36). The cut-off frequency was computed using Newton–Raphson’s algorithm and is approximately 8213 Hz.
(33)$$\begin{array}{cc} \; & h_{roi}=H_{p}\;\cdot \;\frac{h_{\mathrm{res}}}{{\mathrm{FoV}}_{V}}=\frac{h_{\mathrm{res}}}{2}\;\left(\mathrm{pixels}\right) \end{array}$$
(34)$$\begin{array}{cc} \; & w_{roi}=W_{p}\;\cdot \;\frac{w_{\mathrm{res}}}{{\mathrm{FoV}}_{H}}\geq 30\;\left(\mathrm{pixels}\right) \end{array}$$
(35)$$\begin{array}{cc} \; & \mathrm{where}:\;{\mathrm{AoV}}_{H}={\mathrm{AoV}}_{V}\;\cdot \;{\mathrm{aspect}}_{\mathrm{ratio}} \end{array}$$
(36)$$R_{b}=S_{\mathrm{packet}}\;\cdot \;\frac{fps}{2}=\frac{h_{\mathrm{roi}}}{h_{s}\;\cdot \;2}\;\cdot \;\frac{fps}{2}=\frac{h_{\mathrm{res}}}{4}\;\cdot \;\frac{f_{3}\mathrm{dB}\;\cdot \;t_{s}}{2}\;\cdot \;\frac{fps}{2}$$

Taking into account all these starting requirements, three different cases are proposed. Those cases are shown in Figure 5. In the first case, Case I, the camera is perfectly aligned with the shelf’s centre. In Case II, the camera is shifted on the y-axis, and it views the rack from the side. In the last case (Case III), another rack is included, forming a corridor, and the camera is aligned to its centre.

In the remainder of this section, the procedure for calculating the metrics for each case is detailed. In all the cases, the restrictions mentioned above were used to compute the camera’s location by optimising the SUR quantity.

For clarification, Figure 5 shows the geometrical definitions, variables and relations over the scenario’s top and side view. The variables $d_{cam,i}$ are the camera’s relative distance from the shelf, $h_{shelf}$, $l_{shelf}$ the shelf’s height and length, respectively, and $h_{cont}$, $l_{cont}$ the container’s height and length, respectively.

#### 4.1.1. Case I

In this case, to attain the initial restrictions (Equations (Equation 33), (Equation 34) and (Equation 35)), the horizontal FoV, $FoV_{H}$, of the camera must completely cover the entire shelf’s length (Equation (Equation 37)).

The other restriction is that the camera’s vertical FoV, $FoV_{V}$, must be lower than or equal to the shelf’s height (Equation (Equation 38)). Otherwise, the floor and the ceiling will be visible within the image, reducing the photobioreactors’ vertical size.
(37)$$\begin{array}{cc} \; & FoV_{H}=2\;\cdot \;\tan\left(\frac{{\mathrm{AoV}}_{H}}{2}\right)\;\cdot \;d_{\mathrm{cam}},1\geq l_{\mathrm{shelf}} \end{array}$$
(38)$$\begin{array}{cc} \; & FoV_{V}=2\;\cdot \;\tan\left(\frac{{\mathrm{AoV}}_{V}}{2}\right)\;\cdot \;d_{\mathrm{cam}},1\leq h_{\mathrm{shelf}} \end{array}$$

To resolve the stated equations, $AoV_{V}$ must be set to its maximum possible value. As design criteria, it was established as 70${}^{\circ }$. In Section 5, the reason behind not selecting a higher $AoV_{V}$ is discussed.

Using Equations (Equation 37) and (Equation 38), the camera distance, $d_{\mathrm{camera}}$, is obtained, and consequently the SUR. The selected worst-case viewing angle corresponds to the container located further to the right.

#### 4.1.2. Case II

The translation of the camera to the side has the advantage that its distance is considerably reduced. However, as can be seen in Figure 5, the horizontal projection of the reactors shrinks relative to its distance to the camera, and the angle of view also reduces the optical power received by the image sensor. Therefore, Equation (Equation 34) becomes an important restriction. In this case, $FoV_{H}$ is defined in Equation (Equation 39).
(39)$$\begin{array}{cc} FoV_{H} & =2\tan\left(\frac{{\mathrm{AoV}}_{H}}{2}\right)\;\cdot \;d_{{\mathrm{FoV}}_{H}} \end{array}$$
$$\begin{array}{cc} \mathrm{where}: & \;d_{FoV_{H}}=\frac{d_{\mathrm{cam}},1\;\cdot \;\cos\left({\mathrm{AoV}}_{H}/2\right)}{\cos({\mathrm{AoV}}_{H})}\\ \; & \;AoV_{H}=\arctan\left(\frac{l_{\mathrm{shelf}}}{d_{\mathrm{cam}},1}\right) \end{array}$$

In this case, utilizing the maximum $AoV_{H}=70^{\circ }$ restriction, $w_{roi}$ considerably exceeds the restriction imposed in (Equation 34); thus, this configuration is still viable.

We highlight that, in this particular case, the camera does not visualise the whole surface of the nearest containers, but merely a region of their base (as shown in Figure 5). In contrast, the furthest containers are fully scanned. Notwithstanding this uneven configuration, the image projection for each container is preserved (as shown in the example frame). Both the nearest and the furthest containers are projected over the same number of vertical pixels within the image. Therefore, all the containers share the same image area available for communications.

#### 4.1.3. Case III

In this setup, part of the image belongs to the end wall, reducing the available area left for the photobioreactors. In this case, also the horizontal projection constrains the arrangement. If $AoV_{H}$ is too wide, the last reactor’s horizontal size will not reach the minimum imposed (Equation (Equation 34)) and would not be visible within the image. Otherwise, if $AoV_{H}$ becomes too narrow, the first reactor, which is partially visible, could disappear from the image. The size of the horizontal projection for the first container and the last can be computed using Equations (Equation 40) and (Equation 41), respectively.
(40)$$W_{p}^{last}=\frac{d_{\mathrm{cam}},1\;\cdot \;l_{\mathrm{cont}}}{l_{\mathrm{shelf}}+d_{cam,2}-l_{\mathrm{cont}}}$$
(41)$$W_{p}^{first}=\left(l_{\mathrm{shelf}}+d_{cam,2}\right)\;\;\cdot \;\;\left[\tan\left(\frac{{\mathrm{AoV}}_{H}}{2}\right)-\frac{d_{\mathrm{cam}},1}{\left(l_{\mathrm{cont}}+d_{cam,2}\right)}\right]$$

The location of the camera $y_{\mathrm{cam}}$ and dcam is obtained by reducing the horizontal size of both containers (first and last) within the image, attaining the imposed minimum size (Equation (Equation 40)).

### 4.2. Experiments

Two experiments were conducted at the BEA facilities to evaluate signal reception and deterioration due to the increase in biomass concentration. The first experiment evaluated the container surface’s radiance in a controlled indoor environment, whilst in the second experiment, the system’s performance in an outdoor scenario was assessed. The shared materials, resources and equipment are summarised in Table 1.

The following subsections describe each scenario separately to facilitate understanding the motivation, methods and data analysis for each experiment.

#### 4.2.1. Indoor Experiment

In this experiment, the radiance emitted by specific regions from the container’s surface is evaluated. The aim is to determine how the radiance profile changes as the biomass concentration increases for different camera viewing angles. At the same time, the SNR of the channel is measured, and the bit error rate (BER) is estimated. The selected species for this experiment was *BEA 0007B Arthrospira platensis* (*A. platensis*).

Regarding the evaluation methods, the experiment was based on acquiring nine images at different viewpoints with respect to the azimuthal angle ϑ, keeping the link range constant. The radiance for vertical angles was not evaluated because camera translations in the vertical axis were not considered in the theoretical analysis, and for symmetry reasons, a similar behaviour was expected. For each discrete angle, the camera was aligned in the direction to the centre of the container. Figure 6a shows the experimental setup. The biomass concentration was increased in steps of 20 mg l^−1^, starting from 0 mg l^−1^ and reaching 100 mg l^−1^.

After capturing the image samples, they were processed offline to obtain the radiance and SNR, from two different regions, the centre and the boundaries of the container’s surface, to analyse the skewness phenomenon in the outermost regions of the surface.

The radiance was measured indirectly by averaging the pixel values in a tiny window no higher than 50 × 50 pixels. For the area projected in this window and considering the distance and the size of the container, it can be assumed that the radiance pattern varies very smoothly within this region. The original horizontal and vertical window’s length was established at the start of the experiment when $\vartheta =0^{\circ }$. Afterwards, the horizontal size must decrease as a function of the cosine of the angle, ϑ. Consequently, only the radiance contribution of the same original area was evaluated for each angle. The vertical length would remain unaltered as there were no vertical translations. We highlight that because the camera points to the centre of the containers, the windowed area located at the boundaries would decrease not only as a function of the cosine of the view angle, but also as a function of the distance, which varies very slightly for more acute angles. However, the area reduction factor due to this distance increment is considerably lower than the factor due to the view angle, and it can be neglected.

In addition, the SNR was computed within the same window by measuring the mean, $\mu_{\mathrm{chan}}$, and the standard deviation, $\sigma_{\mathrm{chan}}$, of each independent colour channel, Equation (Equation 42), taking into account that there was no other source of light during the experiment. Furthermore, the expected theoretical BER for an on-off keying (OOK) signal can be estimated from Equation (Equation 43) using the complementary error function erfc(·) [3].
(42)$$SNR=20\;\cdot \;log_{10}\left(\frac{\mu_{chan}}{\sigma_{chan}}\right)$$
(43)$$BER=\frac{1}{2}\mathrm{erfc}\left(\sqrt{\frac{SNR}{2}}\right)$$

The key parameters of this experiment are summarised in Table 2. Figure 6b shows some image samples for different biomass concentrations. It can be anticipated by the images obtained that at the borders, the radiance decreases abruptly for internal view angles (in the case of low biomass concentration). This suggests that the initial assumption about the radiance’s skewness phenomenon at the surface’s borders is valid.

#### 4.2.2. Outdoor Experiment

This experiment aimed to evaluate how the presence of external light sources interferes with the signal received by the camera, considering both the light that is reflected on the container’s transmission surface and the light that enters through the sides of the container. In this experiment, the camera node faced just one photobioreactor, which repeatedly transmitted a beacon packet through different biomass concentrations (from low to high). The selected microalgae species was *BEA 1286B Rhodosorus marinus* (*R. marinus*) (brown algae) to evaluate the communication restrictions that imply the use of microorganisms with different absorption curves. The beacon signal is comprised five sequential pulses (green, red, blue, cold white, warm white) followed by a dark guard. The key parameters of this experiment are summarised in Table 3.

## 5. Results

### 5.1. Plant Distribution Study

Table 4 shows the metrics calculated for each case. For illustrative purposes, Figure 7 plots every case in the space defined by the proposed multidimensional metric *F*.

The first configuration, Case I, has a relatively low SUR. Approximately only four percent of the available space is utilised for harvesting. Therefore, this proposal should be discarded, even though the conditions for establishing a communication link are practically ideal. This SUR could be increased if a greater $AoV_{V}$ were selected. Increasing $AoV_{V}$ causes the $FoV_{H}$ to expand, and therefore, the camera can get closer to the shelf, reducing the distance considerably and minimising the required space. However, increasing $AoV_{V}$ without limits would intensify the distortion effects introduced by the lens, bending the vertical lines in the image (barrel distortion), especially for fish-eye lenses. This distortion would not modify the shape of the symbol bands. The symbol bands will always be horizontal (or vertical) straight lines because they result from the RS acquisition method regardless of the lenses’ optics properties. However, it does affect the containers’ shape, reducing its vertical size. Furthermore, this non-linear distortion cannot be mitigated using image processing techniques; doing so would reflect the distortion on the signal bands.

In Case III, however, just the opposite happens. The SUR is almost four times higher than in Case I. However, the optical received power from the most distant containers is so low that it reduces the effective data rate and increases the complexity at reception. This reveals the importance of the camera viewing view. Those acute view angles will significantly affect the total received optical power. They will make it difficult not only to establish the link, but also to estimate the microalgae culture’s biomass precisely.

In conclusion, Case II becomes the optimal choice for this application. It is the solution that achieves a balance in all parameters and has greater replicability. This replicability allows easily creating new plant distributions, such as the one shown in Figure 7. In that example, it is shown how just by adding a second camera on the original shelf and installing a new identical rack that holds the first camera, a corridor similar to Case III can be created. Hence, this configuration makes better use of space at the expense of adding a single low-cost camera. In that case, both the SUR and *N* double their original values, maintaining the communication performance.

However, despite this being the chosen solution, there is still a discussion to be made. The fact that the camera scans only a fraction of the first containers and performs the full scan of those that are further away (due to the image’s viewing perspective) gives rise to new challenges. For example, the signal-to-interference-plus-noise ratio (SINR) is different for each container, especially under the presence of tiny air bubbles in motion within the reactor, which are utilised generally to aerate the organisms and induce a continuous movement. The projection of these small air bubbles increases in size for the nearest containers, generating more noticeable interference phenomena. Furthermore, in terms of biomass sensing, the scanning of just a fraction of the culture can provide reasonable estimations only if the biomass is correctly distributed within the recipient.

On the other hand, analysing the SUR values, it is observed that they are relatively small. The maximum aggregated container’s volume is approximately 13% of the entire required space. However, it is important to notice that, in all vertical cultivation plants, it is necessary to reserve some room for the technicians to execute periodic control routines, handle the extraction of samples with ease and react quickly to warning alarms. Conventionally, it is recommended to reserve up to one meter of separation between racks. This separation coincides with the optimal camera distance in Case II.

### 5.2. Indoor Experiment

Figure 8 shows the intensity radiation pattern emitted from the two selected regions at the container’s surface. The columns represent the area’s location, from left to right: the centre and the border. The rows represent each independent image’s channel, from top to bottom: red, green and blue. Each graph shows the radiation pattern normalised to the maximum radiant intensity for azimuth angles between five and 175 degrees. The radiation patterns for the different concentrations are grouped following a colour gradient from lighter to darker as the concentration increases. Finally, for reference purposes, the Lambertian radiation patterns with n = 1 and n = 2 are shown.

In the container’s central region, the radiation pattern follows a Lambertian with n = 1 for red and green channels when no microorganisms are suspended in the water. In the blue channel, it deviates minimally for wider angles. It can be seen that as the biomass increases, the emission becomes more directive. Actually, in the case of the red and green channel, at the point of maximum concentration, it follows a perfect Lambertian pattern with n = 2. The reason behind this has a connection with *A. platensis*’s SPF and its absorption and scattering cross-section coefficients. *A. platensis* is a planktonic filamentous cyanobacterium. Its cylindrical morphology gives the cell a highly directive SPF [33] (depending on the orientation). The absorption cross-section measured experimentally is moderately high compared to other microalgae organisms.

Therefore, the absorption coefficient, measured experimentally in [25], contributes more to the light extinction than the scattering. This absorption causes the outgoing light at steep angles, which had travelled longer distances than the direct rays, to undergo a significant attenuation, which ultimately produces this directivity on the radiation pattern.

On the other hand, in the blue channel, the radiation intensity decreases faster with the increase in biomass than the red and the green channel. This is to be expected given that the absorption cross-coefficient curve as a function of the wavelength is not flat, but intensifies by almost 1.7 times in the 350–450 nm region of the spectrum [25].

Regarding the emission from the edges, the skewness effect mentioned in Section 3.1 is observed. This emission also becomes more and more directive as the biomass increases.

To better illustrate how the camera perceives the differences in the radiation between both regions (centre and border) at a certain angle and to understand the reason behind selecting these two regions for the analysis, Figure 9 shows some examples of the images captured during this experiment. In this figure, framed in a blue rectangle, the interface that abruptly separates the centre and the border region is highlighted (Figure 9a). Furthermore, this region’s evolution with respect to the increase in concentration is shown in the boxes below.

The location of this interface on the surface depends on the container’s geometric construction and moves with the point of view. In thicker containers, this interface gets closer to the centre, and consequently, the border region increases significantly in size. On the other hand, as the container is viewed from a tighter angle, this interface moves towards the edges (Figure 9c). In this case, the central region extends smoothly towards the borders without reaching them. This interface’s location should be taken into account in the design of the communications link because it delimits two areas with notable differences in light emission that will ultimately impact the SNR. In this setup, the power received from the edges is generally less than from the centre. Therefore, if the sampling occurs at the edges, the signal would be affected by a lower SNR. However, it can be seen that the relative intensity contribution for angles above 30 degrees is higher in the borders than the centre (Figure 8). For these angles, the radiation intensifies relatively. This effect is also observed in Figure 9b, where the edge region appears brighter than the centre. This outcome is related to the SPF of the *A. platensis* and the internal shape of the container.

Regarding the SNR, Figure 10 shows the SNR for different viewpoints either when the signal is received from the centre (dashed lines) or the side (dotted lines). The columns represent the biomass concentration, increasing from left to right. The rows represent the image channel, from top to bottom: red, green and blue. Each graph shows the SNR against the camera viewing angle (from zero to 85 degrees).

The most evident result extracted from these graphs is that the SNR in the blue channel decreases much faster than in the other channels. Furthermore, the SNR starts to decay at 40 degrees at the edges, approximately 20 degrees earlier than in the red and green channels. This was discussed previously in terms of the light intensity emitted by the surface. Nevertheless, another result can be extracted. The differences between the central and outer SNRs are reduced progressively as the biomass concentration increases. *A. platensis* scattering slowly contributes to distributing the optical signal more evenly across the surface. Besides, the difference in power also gradually decreases, at least in the red and green channels. Therefore, slow attenuation combined with increasing scattering is beneficial, mainly because it reduces the gap between the two regions, increasing the effective signal reception area. Finally, regarding the sensing of the biomass concentration and culture’s growth state, it is concluded that it slightly depends on the viewing angle. The relative differences in attenuation per angle for each channel as a function of concentration are non-linear. As the concentration increases, the radiant intensity diagram varies slightly in shape, relatively different for each channel. Therefore, to estimate the biomass correctly using only the pixel values inside a given area, it is necessary to consider the viewpoint. However, this radiation’s behaviour provides valuable information that can be favourably exploited for more accurate crop parameters’ estimation. The larger the variations, the better the estimation will be. In this sense, Case II, which proved to be the ideal candidate for a real deployment, has the added advantage that it analyses the strain from different angles. If all containers are interconnected through pipes and the strain is transferred from container to container periodically, its sensing will be more accurate.

### 5.3. Outdoor Experiment

The results are presented in Figure 11. This figure displays a snapshot taken for three concentration densities: low (a), medium (b) and high (c). The white rectangle encloses the detected beacon signal. The graphs shown on the picture’s right side represent the red, green and blue pixel values from a one-pixel column located within this rectangle. Furthermore, Figure 11d represents the reference case where the signal is extracted without being affected by the microalgae channel. The signal recovered corresponds to the sequentially pulsed LEDs: white cold, white warm, dark guard (no pulse), green, red and blue.

It can be observed that this species produces significant attenuation in the blue and green portions of the spectrum (attaining the Bayer filter’s spectrum response of the camera). Despite that the green, blue and red pulses cannot be distinguished in Figure 11c (due to low optical transmitting power), it is still possible to differentiate the white cold and white warm spectral signatures. Therefore, those LEDs are viable for communication purposes.

Finally, analysing the reference signal (Figure 11d), it is observed that the sunlight increases the light power received at the container’s surface. It sets an offset value and consequently reduces the dynamic range available for communications. However, as the concentration increases, this offset value decreases because of the attenuation of the microalgae. This implies that the external light that enters through the container’s sides is attenuated, just like the signal of interest. Therefore, in high biomass concentration conditions, increasing the photobioreactor’s light emission power will expand the available dynamic range. Pixel values obtained from dark pulses will tend gradually to zero, while the signal power is increased on purpose.

## 6. Conclusions

In this work, OCC was proposed as a suitable communication technology for monitoring microalgae production plants based on artificial lighting. The theoretical channel model was introduced alongside the analysis of the parameters that significantly impact the achievable data rate, such as the nodes’ geometrical configuration and the camera sampling frequency and exposure time.

This research highlighted the importance of optimising plant distribution in terms of link quality, channel capacity and efficient space exploitation. A multidimensional metric was designed for this purpose and tested in the conducted case study, where three different node arrangements were classified. This study’s main result was that the configuration in which the camera observes the reactor shelf from the side is the preferred solution. On the one hand, it provides comparable link qualities between all containers without incurring detrimental losses of the signal strength due to the camera’s viewing angle. On the other hand, it better exploits the scarce space available than a camera watching the containers frontally.

In addition, the experimental evaluation of the proposed flat-panel photobioreactor prototype was carried out in indoor and outdoor environments for two different microalgae species: *A. platensis* and *R. marinus* (green and brown algae, respectively). In the indoor experiment, the container surface’s radiation pattern was measured for each image channel at different concentrations. This analysis reveals significant differences in the SNR between channels due to the algae absorption spectrum. Moreover, notable differences can be observed between the borders and the central part of the surface. These differences tend to reduce as the concentration of microalgae increases due to the phenomenon of scattering. Despite that the viewpoint must be taken into account when designing the system, it additionally provides valuable information for in situ sensing the algae’s biomass and its growth state. In the outdoor experiment, the blue and green channels were highly attenuated, which compelled discarding these wavelengths for data transmission. Furthermore, in this scenario, the sunlight decreased the available dynamic range considerably for signal transmission, especially for low biomass concentration. Finally, examining the obtained results for the two microalgae species with different absorption profiles determined that the selected strain and its temporal evolution must be considered in the development of the optical link.

Future research should further develop and confirm these initial findings by extending the SNR’s evaluation to the actual achievable BER for this deployment in both indoor and outdoor scenarios. It should examine more deeply how the presence of aeration bubbles might affect the signal reception. It could also focus on comparing and developing novel framing strategies for sensor data, reducing packet overhead, easing transmitter discovery and data synchronisation and equalisation. In terms of culture sensing, artificial intelligence for biomass estimation may constitute the object of future studies.

In conclusion, it was demonstrated that microalgae production plants are a potential use case for OCC. This technology provides simultaneous node monitoring capabilities using a cost-effective deployment, paving the way to develop smart farming strategies within the microalgae cultivation field.

## Figures and Tables

**Figure 1 sensors-21-01621-f001:**
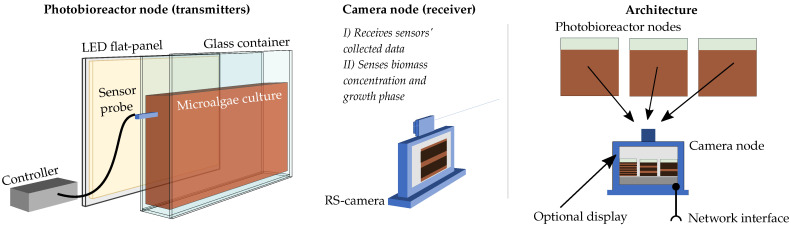
Proposed architecture based on flat-panel photobioreactors. RS, rolling-shutter.

**Figure 2 sensors-21-01621-f002:**
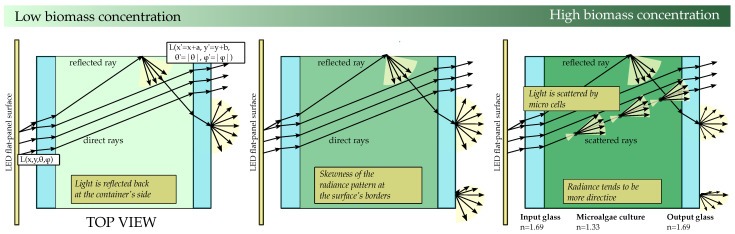
Layered version of the channel.

**Figure 3 sensors-21-01621-f003:**
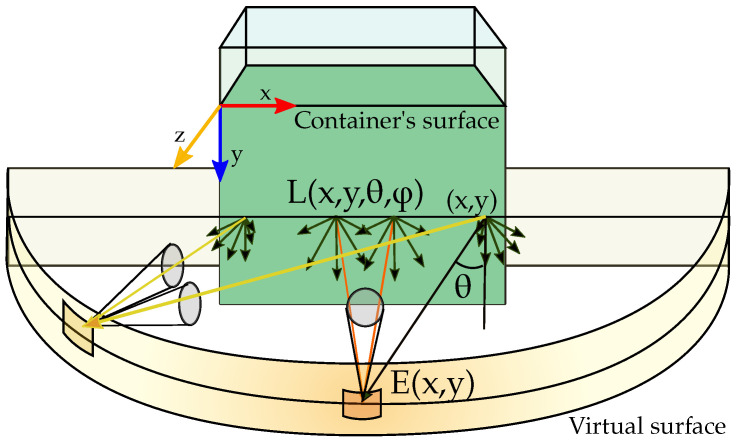
Virtual surface that encloses the container.

**Figure 4 sensors-21-01621-f004:**
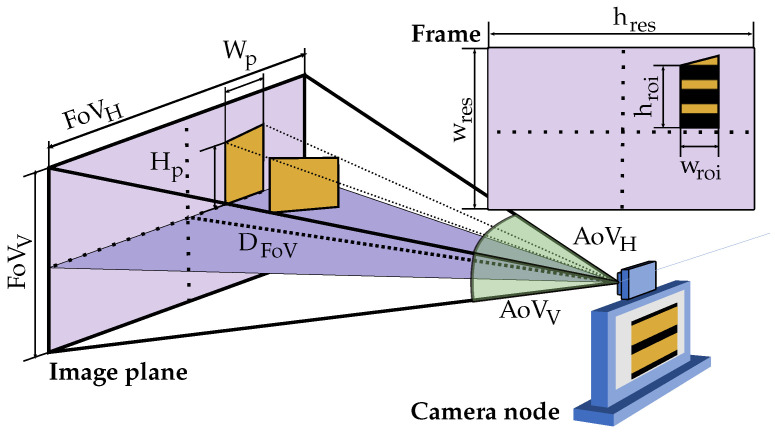
Geometrical parameters involved in the computation of the maximum achievable node’s data rate.

**Figure 5 sensors-21-01621-f005:**
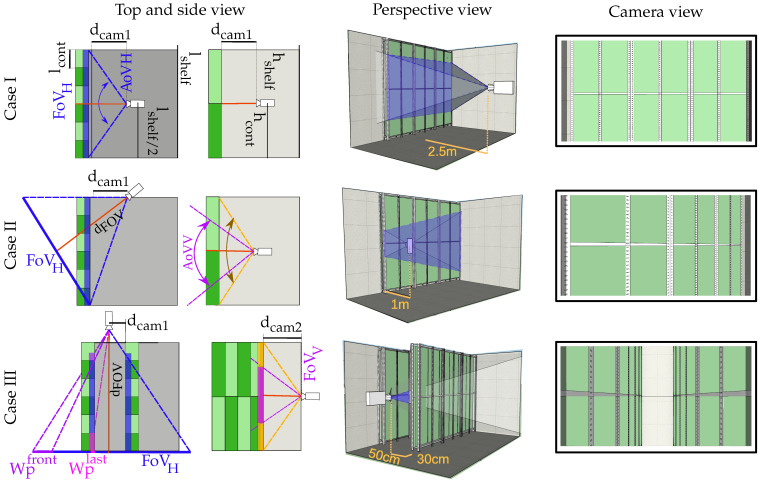
Three different cases proposed for the plant distribution study.

**Figure 6 sensors-21-01621-f006:**
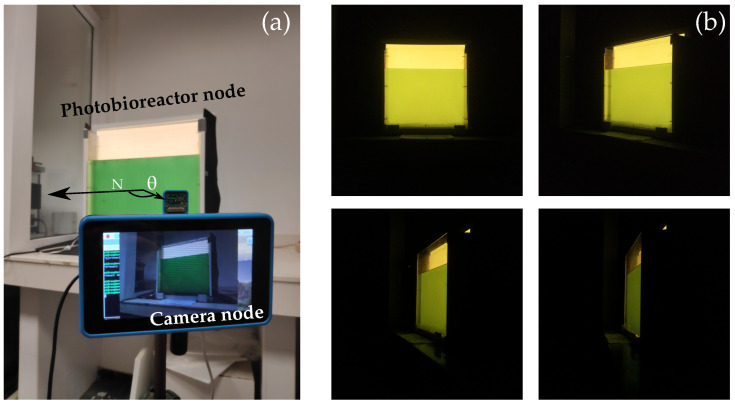
Pictures of the indoor experiment. (**a**) shows the experimental setup, and (**b**) depicts exemplary images captured at different angles for different biomass concentrations.

**Figure 7 sensors-21-01621-f007:**
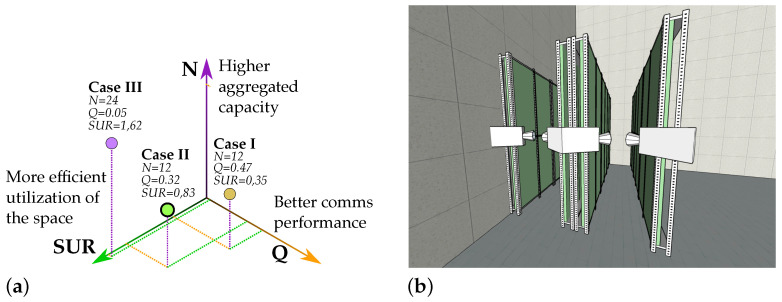
Illustration of the results of the plant distribution study. (**a**) Space generated by the multidimensional metric F; (**b**) example of Case II’s replication process.

**Figure 8 sensors-21-01621-f008:**
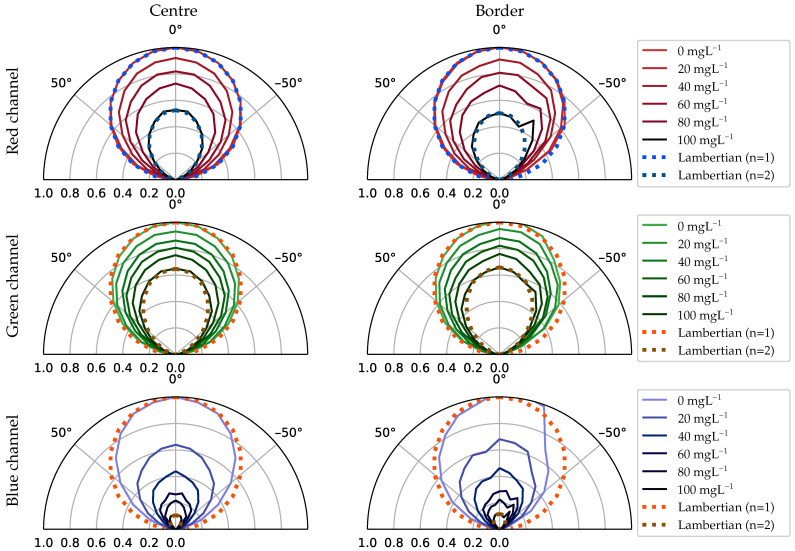
Red, green and blue radiation diagram of small areas located at the centre and the border of the container.

**Figure 9 sensors-21-01621-f009:**
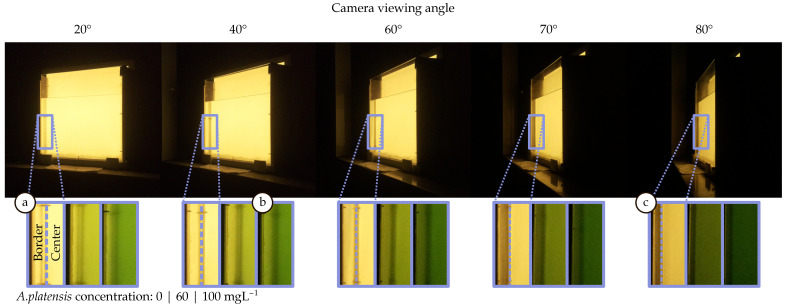
Illustrative example of the results obtained in the indoor experiment.

**Figure 10 sensors-21-01621-f010:**
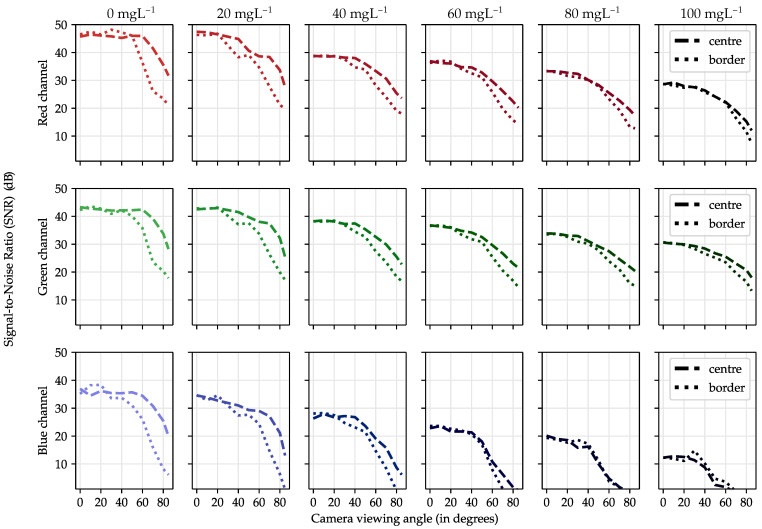
Red, green and blue SNR for small areas located at the centre and the border of the container.

**Figure 11 sensors-21-01621-f011:**
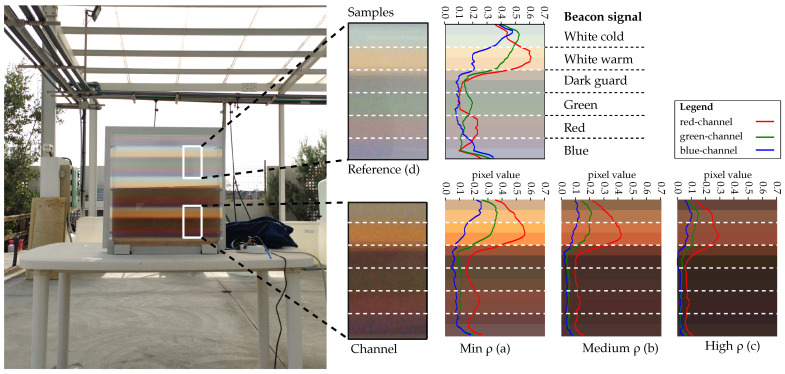
Experimental transmission of a well-known beacon signal.

**Table 1 sensors-21-01621-t001:** Resources and equipment. * (wavelengths (nm): 630 (red).

Photobioreactor Node		Camera Node	
**Part**	*Parameters*	**Part**	*Parameters*
**LED Lamp**	Eglo Tunable White - RGB connect	**Camera**	PiCamera Version 2
	- 1 white cold LED (6500K)		- Image sensor: Sony IMX586 [32]
	- 1 white warm LED (2700K)		- Aperture lens: f/2
	- 1 RBG LED *		- Focal length (equivalent) (mm): 3
			- Image resolution (px): 3280 × 2464
	530 (green), 475 (blue))		- Sampling time $t_{s}$ (*s*): 18.904
**Container**	Square glass panels (custom)	**Receiver**	Raspberry Pi 3 Model B
	- Dimensions (cm) : 50 × 50 × 9		

**Table 2 sensors-21-01621-t002:** Indoor experiment key parameters.

Parameter	*Value*
Horizontal view angle (ϑ)	$0^{\circ }$ to $80^{\circ }$ in steps of 10 degrees
Distance	1.5 m
LEDs	Warm and cold white LEDs
Camera	PiCamera version 2
Microalgae	
Genus	Arthrospira
Species	*BEA 0007B Arthrospira platensis*
Biomass concentration (mg l^−1^)	0 to 100 in steps of 20

**Table 3 sensors-21-01621-t003:** Outdoor experiment key parameters.

Parameter	*Value*
Optical signal	Beacon
Chip duration $T_{\mathrm{chip}}$	1/8400
Distance (m)	2
Microalgae	
Genus	Rhodosorus
Species	*BEA 1286B Rhodosorus marinus*
Biomass concentration (mg/L)	(estimated) 75, 195, 430
Camera	
Model	Mi 9T Pro
Image sensor	Sony IMX586
Image resolution (px)	4000 × 3000
Focal length (mm)	4.8 mm
Aperture value	F/1.7
ISO speed rating	450, 490, 670
Exposure time (s) $t_{\exp}$	100

**Table 4 sensors-21-01621-t004:** Metric values obtained for the three arrangements.

Case	$V_{\mathrm{room}}$ (m${}^{2}$)	$V_{\mathrm{cont}}$ (m${}^{2}$)	$h_{\min}$	L(ϑ,φ)
I	23.31	0.81	0.5	0.95
II	9.81	0.81	0.5	0.65
III	11.68	1.62	0.5	0.12
**N**	**SUR**		**Q**	
12	0.035		0.47	
12	0.083		0.32	
24	0.139		0.05	

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
