# Peer review of "Optical Camera Communication as an Enabling Technology for Microalgae Cultivation†"

_sensors, 2021, doi:10.3390/s21051621_

Round 1

Reviewer 1 Report

The work Optical Camera Communication as an enabling technology for microalgae cultivation presents  a system developed for  microalgae  production plant.

The OCC proposed a represent a great communication technology for monitoring  microalgae production plants based on artificial lighting. I believe that the proposed technology represents an important step in the technology of cultivation of microalgae, helping a real-time monitoring of the microalgae culture, thus avoiding the accidents to which cultures are predisposed and at the same time considerably diminishing the work of the operators. At the same time, the application of the proposed technology will lead to a decrease in the production costs of microalgae biomass.

Reviewer 2 Report

Dear Authors,

you presented interesting and novel idea which I found valuable. For that reason I'd like to recommend it for publication in Sensors Journal.

Best regards

Reviewer 3 Report

The paper describes the use of OCC for microalgae production plants. The authors focus on how the communication capabilities can be associated / affected with the microalgae production and the physical configuration of the plant. Experiments were made with 2 species of microalgae, several camera dispositions and microalgae concentrations, and indoor/outdoor.

The paper topic is very interesting and it requires a real multidisciplinary approach. I did not found any significant problems with the paper or its writing. It has relevant and updated references, introduction and methods are adequately described and it has promising results. Another strong point of the paper is the use of figures. They are really nice and help the understanding of the text. 

I will only make some small suggestions to improve a lit bit more its quality.

=====

Introduction: the paper scientific contribution could have a better highlight in this section. It could also have a paragraph detailing how useful can be a OCC in practice (it can improve the reader interest): Why would I choose to use it instead of another technology? What kind of useful data can I transfer? 

Section 3 and 4: I understand the use of SNR, but one thing that I missed (or it was not clear for me) was an actual transmission of data and its impact under these SNR. In other words, how much data can I transfer under these circumstances and how much is acceptable for this application? What is the minimum sampling time for the microalgae production information update? 

  • Although this is not critical, it can improve the understanding of the real impact of the results.

Results: The outdoor experiment is a little bit off with the rest of the results.. especially when we read Line 30 in the beginning of the paper that states the problems of the open air plants. I do not suggest to take it off, but maybe a small rewriting of this part to better connect/allign with the rest of the paper.

- Figure 11: please add the axes name/units.

Conclusion: it is missing a "future work" text by the end.

Reviewer 4 Report

The investigation of optical camera communication as an enabling technology for microalgae cultivation is of great importance. The authors have carried out sufficient theoretical and experimental research in this work. It is recommended to be published in Sensors. However, language editing is necessary for the manuscript. For example:

“In this work, both a preliminary study of the plant distribution for a real case is carried out and

the system’s experimental evaluation in terms of communications.”

“Highlight that In Figure 5 it can be seen that for the nearest containers, the camera only examines

their bottom part, contrary to the furthest ones, which are observed entirely.”

“In all cases, It is decisive to minimise this term to design a viable and competitive solution that meets the associated production costs.”
